# Prevalence and determinants of food insecurity among pregnant women in Nigeria: A multilevel mixed effects analysis

Otobo I. Ujah[1,2]*, Pelumi Olaore[2], Chukwuemeka E. Ogbu[2], Joseph-Anejo Okopi[3], Russell S. Kirby[2]

1 Department of Obstetrics and Gynaecology, Federal University of Health Sciences, Otukpo, Nigeria,
2 College of Public Health, University of South Florida, Tampa, Florida, United States of America,
3 Department of Microbiology, Federal University of Health Sciences, Otukpo, Nigeria

* otoboujah@yahoo.com

## Abstract

Food insecurity (FI) remains a key priority for sustainable development. Despite the well-known consequences of food insecurity on health and well-being, evidence regarding the burden and determinants of FI among pregnant women in Nigeria is limited. Framed by the social-ecological model, this study aimed to determine the prevalence of FI, and its associations with individual-/household-level and contextual-level factors among pregnant women in Nigeria. A cross-sectional study based on the Nigerian Multiple Indicator Cluster Survey (2021 Nigerian MICS6) was conducted among a sample of 3519 pregnant women aged 15–49 years. Several weighted multilevel multinomial logistic regression models were fitted to assess the association between individual-/household-s level and community-level characteristics with FI. We estimated and reported both fixed effects and random effects to measure the associations and variations, respectively. Results: The prevalence of FI among pregnant women in Nigeria was high, with nearly 75% of the participants reporting moderate to severe FI in the past 12 months (95% CI = 71.3%-75.8%) in 2021. There were also significant differences in all the experiences of food insecurity due to lack of money or resources, as measured by the Food Insecurity Experience Scale (FIES), except for feeling hungry but not eating because of lack of money or resources ($p < 0.0001$). Multivariate analysis revealed that higher parity, households with 5 or more members, household wealth index, urban residence, and community-level poverty were significantly associated with FI. Our study demonstrates a significantly high prevalence of FI among pregnant women in Nigeria in 2021. Given the negative consequences of FI on maternal and child health, implementing interventions to address FI during pregnancy remains critical to improving pregnancy outcomes.

**Data Availability Statement:** The data set is publicly available and can be downloaded from https://mics.unicef.org/surveys.

**Funding:** The authors received no specific funding for this work.

**Competing interests:** The authors have declared that no competing interests exist.

## Introduction

Ensuring food security globally is a key priority for health, social and economic development. Food security is achieved when individuals have access to, and are able to afford, safe and nutritious foods that meet their dietary requirements and preferences for optimal health and well-being [1]. Several measures have been implemented across different parts of the world to address the growing burden of food insecurity (FI). For example, in Nigeria, endeavors to combat food insecurity encompass initiatives such as the National Accelerated Food Production, Operation Feed the Nation, Agricultural Development Programme, Structural Adjustment Programme, National Poverty Eradication Programme (NAPEP), and more [2]. However, these interventions have acheived limited levels of success. In the United States, such initiatives have included programs such as the Supplemental Nutrition Assistance Program (SNAP) and the Special Supplemental Nutrition Program for Women, Infants, and Children (WIC) [3]. Similarly, in Ethiopia, strategies such as the food security package (FSP) program and the food-for-work (FFW) program have been implemented to combat household food insecurity. Despite concerted national and global efforts to alleviate food insecurity, it remains an ongoing public health crisis [4].

According to estimates by the United Nations Food and Agriculture Organization (UN FAO), approximately 2.4 billion people experienced moderate or severe FI in 2022 [5]. This burden was exceptionally high in sub-Saharan Africa, where about 67.2% of individuals experienced moderate or severe FI and 24.0% experienced severe food insecurity. Figs 1 and 2 depict the trends in the prevalence of FI (moderate/severe and severe) in sub-Saharan Africa and Western Africa between 2014–2022 based on data from the UN FAO [5].

Women, particularly during the peripartum period, are disproportionately susceptible to FI. This vulnerability can be attributed to increased nutritional demands, challenges associated with food preparation, and financial strain arising from leaving the workforce, especially in the postpartum period [6, 7]. It is also noteworthy that while maternal preconception and prenatal nutritional status, which is often suboptimal in low- and middle-income countries (LMICs), significantly predicts adverse pregnancy outcomes [7, 8] and is consequently likely to be affected by FI, relatively little attention has been given to addressing food FI as a crucial aspect of women's well-being [9].

Food insecurity during pregnancy has been linked to various adverse maternal and perinatal outcomes, including psychological effects such as anxiety, depression, and stress, as well as cardiometabolic effects like increased weight gain and gestational diabetes [10–12]. Moreover, higher levels of FI have been associated with an increased risk of birth defects such as cleft palate, congenital heart diseases, and neural tube defects [10]. A plausible mechanism for the observed associations are driven by stress resulting from inadequate food consumption and nutrient deficiencies which in turn results in perinatal complications [11]. In resource-constrained settings, these direct and indirect negative consequences of FI are likely to be worse for pregnant women. Moreover, the COVID-19 pandemic has likely exacerbated FI in sub-Saharan Africa, where resources are distributed unequally [9, 13].

In spite of the well-established impact of FI on pregnancy outcomes, there is a lack of nationally robust evidence regarding the magnitude and determinants of food insecurity among pregnant women in Nigeria. Although a study conducted in Nigeria found higher prevalence of FI among pregnant women in rural areas compared to urban areas [14], it is important to note that this study is fraught with several limitations such as a small sample size, lack of national representativeness, and a failure to account for individual- and contextual-level variables that contribute to food insecurity among pregnant women. These gaps in the existing literature present challenges in designing effective innovations and evidence-based strategies

**Fig 1. Trends in the prevalence of food insecurity (moderate/severe and severe) in sub-Saharan Africa, 2014–2022 (Data points from FAO, IFAD, UNICEF, WFP and WHO. 2022.** *The State of Food Security and Nutrition in the World 2022. Transforming food systems for food security, improved nutrition and affordable healthy diets for all.* Rome, FAO).

and policies that could enhance pregnancy outcomes, as well as maternal and child health and well-being.

Therefore, this study aims to extend existing knowledge regarding the burden of FI among pregnant women in Nigeria by leveraging nationally representative data. Specifically, this study aimed to answer the following research questions:

1. What is the prevalence of FI among pregnant women in Nigeria? What are the factors that predict the likelihood of being at or below a FI level?

2. How do individual and household characteristics influence the probability of being at or below a FI level for each category?

By addressing these questions, this study potentially contributes to our knowledge of FI in Nigeria and informs strategies to improve maternal, fetal and child health outcomes.

## Theoretical framework

In this study, we used Bronfenbrenner's social-ecological model (SEM) as the framework for guiding our understanding of the complex interplay of factors associated with FI among pregnant women in Nigeria. This model posits that FI is inherently shaped by a multiple factors spanning various levels of influence, encompassing microsystem (intrapersonal), mesosystem (interpersonal), exosystem (community), and macrosystem (policy) levels of influence [15–17]. The model describes how interactions among factors within and across each level culminate in a specific outcome at the individual or microsystem level of influence [18]. The

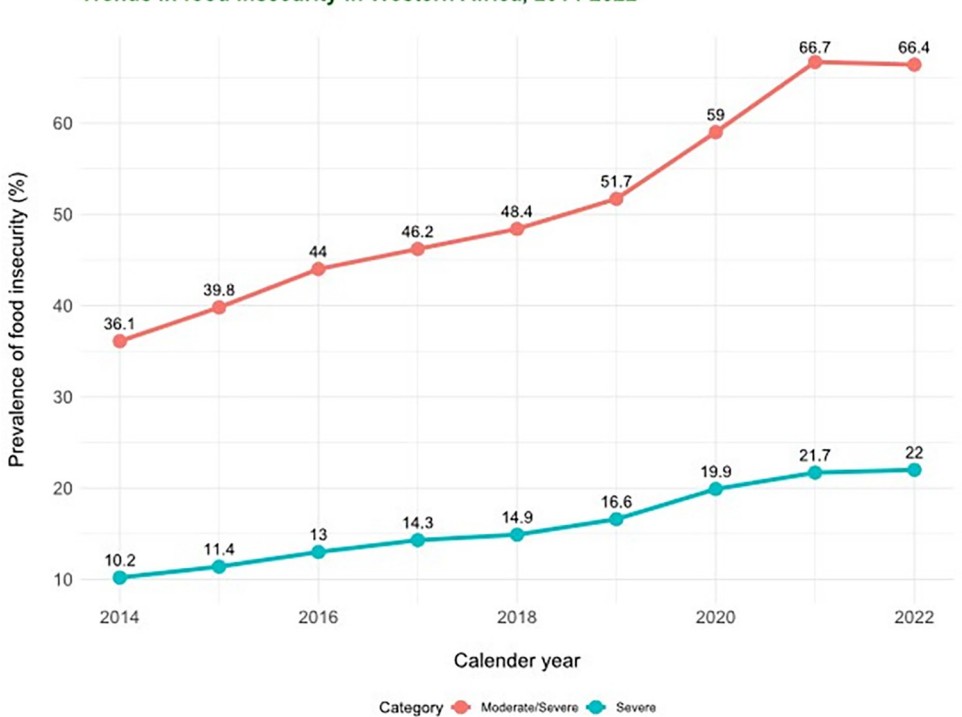

**Fig 2. Trends in the prevalence of food insecurity (moderate/severe and severe) in West Africa, 2014–2022 (Data points from FAO, IFAD, UNICEF, WFP and WHO. 2022.** *The State of Food Security and Nutrition in the World 2022. Transforming food systems for food security, improved nutrition and affordable healthy diets for all.* Rome, FAO).

hierarchical nature of this model formed the basis for which explanatory variables and the empirical strategy were chosen and operationalized. Previous studies have utilized the SEM to investigate correlates of food insecurity across a diverse range populations and contexts [19–22].

## Methods

### Ethics statement

The Nigeria MICS procedures were reviewed and approved by the National Bureau of Statistics (NBS) and UNICEF. According to the 2021 MICS6 report, all participants provided verbal consent before the administration of questionnaires. In the case of participants under 18 years (minors), informed consent was obtained from their parents or legal guardians. Participants were assured of voluntary participation, confidentiality, the anonymity of their information, and the freedom to withdraw from the interview at any point. The data analyzed in this study was acquired from UNICEF through a formal request. As this study involved secondary analysis of publicly available de-identified Nigeria 2021 MICS6 data, it was deemed exempt from the human subject research approval process.

### Study design and data source

This study is based on quantitative cross-sectional data derived from the Nigeria 2021 Multiple Indicator Cluster Survey (MICS6), which is a nationally representative survey that collects sociodemographic and health indicators from both males and females aged 15–49 years. The survey utilized a multistage stratified cluster sampling approach that employed a probability

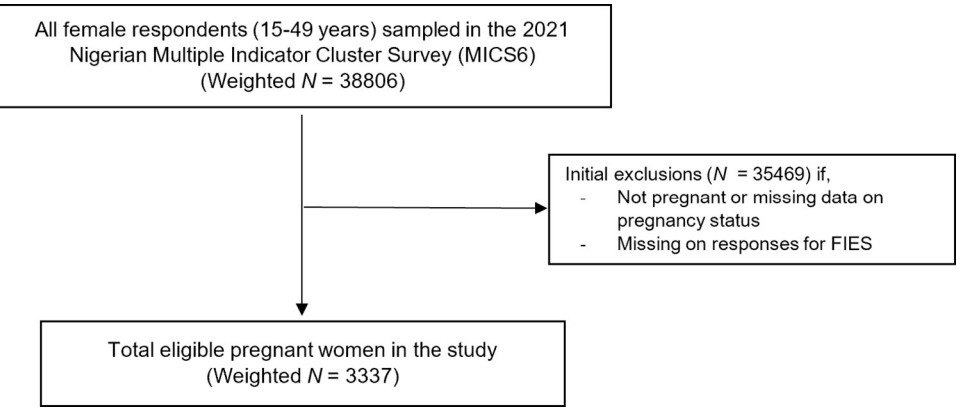

**Fig 3. Shema representing sample selection from the MICS6 data.**

proportional to size to select enumeration areas in the first stage based on the 2006 Population and Housing Census of the Federal Republic of Nigeria (NPHC). In the second stage, 20 households were randomly selected within each enumeration area. If multiple eligible pregnant respondents were present within a household, only one respondent was chosen randomly for the interview. Data were collected using Computer-Assisted Personal Interviewing (CAPI) technology through face-to-face interviews with respondents in their respective households. The survey report provides more detailed information on the sampling design and data collection methods. The survey data are publicly available and can be accessed from https://mics. unicef.org/surveys. In order to account for the complex survey design (i.e. weighting, clustering, and stratification), we used the national women's sample survey weights for reporting survey results.

## Sampling and study population

This study was limited to a sub-sample of participants from the Nigerian MICS6 survey, which included a representative sample of pregnant women aged 15–49 years who provided information on their household's FI experience at the time of completing the survey (*n* = 3565 respondents). We excluded observations with missing or "Don't know" responses on FI experience from the analysis (*n* = 46), resulting in a final analytical unweighted sample of 3519 pregnant women (Weighted *N* = 3337) nested in 1318 primary sampling units (Fig 3).

## Measures

### Dependent variable

The outcome of this study was food insecurity. As shown in Table 1, the MICS measures household FI using the standardized eight-item Food Insecurity Experience Scale (FIES). The FIES was developed by the United Nations Food and Agriculture Organization (UN FAO) to provide internationally comparable estimates of the magnitude of FI experience in accordance with the Sustainable Development Goal (SDG) indicator 2.1.2 - "*prevalence of moderate or severe FI in the population, based on the Food Insecurity Experience Scale (FIES)*" [23]. Food insecurity was assessed based on the respondents' recall of their experiences of food insecurity within their household over the past 12 months. The possible responses in each question in the FIES include "*Yes*" scored as 1, "*No*" scored as 0 and "*Don't know*" scored as DK. Raw household FI scores (0–8) were computed as the total number of affirmative responses, and participants were categorized into three levels of FI based on their scores. Scores ranging from 0 to 3

**Table 1. English version of the Food Insecurity Experience Scale (FIES).**

| N. | Short reference | Question |
|---|---|---|
| 1 | WORRIED | During the last 1 year, was there a time when you were worried you would not have enough food to eat because of a lack of money or other resources? |
| 2 | HEALTHY | During the last 1 year, was there a time when you were unable to eat healthy and nutritious food because of a lack of money or other resources? |
| 3 | FEWFOODS | During the last 1 year, was there a time when you or others in your household ate only a few kinds of foods because of a lack of money or other resources? |
| 4 | SKIPPED | During the last 1 year, was there a time when you or others in your household had to skip a meal because there was not enough money or other resources to get food? |
| 5 | ATELESS | During the last 1 year, was there a time when you or others in your household ate less than you thought you should because of a lack of money or other resources? |
| 6 | RANOUT | During the last 1 year, was there a time when your household ran out of food because of a lack of money or other resources? |
| 7 | HUNGRY | During the last 1 year, was there a time when you or others in your household were hungry but did not eat because there was not enough money or other resources for food? |
| 8 | WHOLEDAY | During the last 1 year, was there a time when you or others in your household went without eating for a whole day because of a lack of money or other resources? |

indicate food secure, scores ranging from 4 to 6 indicate moderate FI, and scores ranging from 7 to 8 indicate severe FI based on previous literature [24].

## Predictor variables

The explanatory variables used in the analysis were selected based on a comprehensive review of the literature, their biological plausibility in the exposure-outcome relationship, and their availability in the survey data studied (Table 2). These variables were classified into individual-, household-, and community-level factors based on the social-ecological model framework. Individual-level variables included maternal age (15–24 years, 25–34 years, and 35+ years), married or cohabiting (yes vs. no), birth events (0, 1–2, 3–4, and 5+), health insurance status (insured and not insured), household wealth index (poorest, poor, middle, wealthier, and wealthiest), and pregnancy intention (planned and unplanned). Household-level variables included household members (<5 and >5), religion (Christianity and non-Christian), and children under 5 years (none, 1, and 2+).

The MICS survey data did not directly collect community-level factors such as poverty, except for place of residence (urban and rural) and region (North-Central, North-East, North-West, South-East, South-South, and South-West). To account for community-level poverty, we derived it by aggregating household-level poverty within their respective clusters and classified it as low, middle, and high.

## Statistical analysis

We used SAS version 9.4 (SAS Institute, Cary, North Carolina, USA) and R version 4.2.2 for all our statistical analyses. In order to provide valid estimates of the standard errors based on the complex survey design, we calculated weighted means, prevalence estimates, and confidence intervals for maternal characteristics and FI status using SAS procedures PROC SURVEYMEANS for continuous variables and PROC SURVEYFREQ for categorical variables.

To assess the associations between the predictor variables and FI, we conducted bivariate analyses using the Chi-squared test. Additionally, we conducted a preliminary check for multicollinearity before constructing our multilevel models. While the diagnostic results for multicollinearity are not presented in the paper, it is important to note that we did not find any

**Table 2. Characteristics of the study population by food security status among pregnant Nigerian women aged 15–49 years, MICS6[a].**

| | Study population | | Experiences of FI | | | | | | | | | |
| | | | Food secure | | | Moderate FI | | | Severe FI | | | |
| | Weighted | | Weighted | | | Weighted | | | Weighted | | | |
| Variable | N | % | n | % | 95% CI | n | % | 95% CI | n | % | 95% CI | $\chi^2$ |
|---|---|---|---|---|---|---|---|---|---|---|---|---|
| Overall | 3337 | 100 | 882 | 26.4 | 24.2-28.7 | 996 | 29.9 | 27.9-31.9 | 1459 | 43.7 | 41.3-46.1 | <0.0001*** |
| **Individual-level factors** | | | | | | | | | | | | |
| Age group (y) | | | | | | | | | | | | |
| 15-24 | 962 | 28.9 | 294 | 30.6 | 26.7-34.7 | 277 | 28.2 | 25.3-32.6 | 391 | 40.6 | 36.6-44.8 | **10.0*** |
| 25-34 | 1534 | 46.0 | 392 | 25.6 | 22.5-28.9 | 480 | 31.3 | 28.3-34.4 | 662 | 43.2 | 40.1-46.3 | |
| 35+ | 840 | 25.2 | 195 | 23.2 | 19.0-28.1 | 239 | 28.5 | 24.8-32.4 | 406 | 48.3 | 43.7-52.9 | |
| Married/cohabiting status | | | | | | | | | | | | |
| No | 110 | 3.3 | 20 | 18.0 | 9.9-30.3 | 35 | 32.3 | 20.0-47.6 | 55 | 49.8 | 36.9-62.8 | 1.9 |
| Yes | 3227 | 96.7 | 862 | 26.7 | 24.4-29.1 | 961 | 29.8 | 27.8-31.8 | 1404 | 43.5 | 41.2-45.9 | |
| Parity | | | | | | | | | | | | |
| 0 | 505 | 15.1 | 181 | 36.0 | 30.4-42.0 | 127 | 25.1 | 20.2-30.8 | 196 | 38.9 | 33.0-45.1 | **30.4*** |
| 1-2 | 1120 | 33.6 | 318 | 28.4 | 24.9-32.1 | 336 | 30.0 | 26.7-33.5 | 466 | 41.6 | 37.9-45.5 | |
| 3-4 | 831 | 24.9 | 214 | 25.8 | 21.4-30.7 | 259 | 31.1 | 27.6-34.8 | 358 | 43.1 | 38.7-47.6 | |
| ≥5 | 881 | 26.4 | 168 | 19.1 | 16.1-22.4 | 275 | 31.2 | 27.3-35.4 | 438 | 49.8 | 45.4- 54.1 | |
| Health insurance coverage | | | | | | | | | | | | |
| No | 3260 | 97.7 | 855 | 26.2 | 24.1-28.5 | 967 | 29.7 | 27.7-31.7 | 1438 | 44.1 | 41.7-46.5 | 0.2 |
| Yes | 77 | 2.3 | 27 | 35.0 | 18.1-56.8 | 29 | 37.5 | 22.9-54.8 | 21 | 27.4 | 15.0-44.7 | |
| Pregnancy intention | | | | | | | | | | | | |
| Planned | 2600 | 77.9 | 682 | 26.2 | 23.7-28.9 | 783 | 30.1 | 27.9-32.4 | 1135 | 43.7 | 41.0-46.3 | 0.7 |
| Unplanned | 731 | 21.9 | 197 | 26.9 | 22.1-32.3 | 213 | 29.1 | 24.7-33.9 | 322 | 44.0 | 39.1-49.1 | |
| **Household-level factors** | | | | | | | | | | | | |
| Household members | | | | | | | | | | | | |
| ≤ 5 | 1219 | 36.5 | 417 | 34.2 | 30.3-38.3 | 343 | 28.1 | 24.9-31.6 | 459 | 37.7 | 34.0-41.5 | **33.2*** |
| > 5 | 2118 | 63.5 | 465 | 21.9 | 19.5-24.6 | 653 | 30.8 | 28.4-33.4 | 1000 | 47.2 | 44.4-50.1 | |
| Religion | | | | | | | | | | | | |
| Christian | 1150 | 34.4 | 289 | 25.2 | 21.1-29.7 | 331 | 28.8 | 25.5-32.4 | 529 | 46.0 | 41.4-50.6 | 1.7 |
| Non-Christian | 2187 | 65.5 | 592 | 27.1 | 24.5-29.7 | 664 | 30.4 | 28.1-32.8 | 930 | 42.5 | 39.9-45.2 | |
| Children under 5 in household | | | | | | | | | | | | |
| None | 879 | 26.3 | 259 | 29.5 | 25.3-33.9 | 249 | 28.3 | 24.1-32.8 | 371 | 42.3 | 37.6-47.1 | 0.5 |
| 1 | 1342 | 40.2 | 344 | 25.7 | 22.3-29.3 | 417 | 31.1 | 28.0-34.4 | 580 | 43.2 | 39.8-46.8 | |
| 2+ | 1116 | 33.4 | 278 | 24.9 | 21.7-28.5 | 330 | 29.6 | 26.3-33.1 | 507 | 45.5 | 41.6-49.4 | |
| Household wealth index | | | | | | | | | | | | |
| Poorest | 845 | 25.3 | 176 | 20.8 | 17.3-24.8 | 247 | 29.2 | 25.6-33.1 | 422 | 50.0 | 45.8-54.2 | **72.1*** |
| Poorer | 795 | 23.8 | 193 | 24.3 | 21.2-27.8 | 255 | 32.1 | 28.6-35.9 | 346 | 43.5 | 39.4-47.8 | |
| Middle | 660 | 19.8 | 153 | 23.2 | 19.4 27.5 | 198 | 30.0 | 25.1-35.4 | 308 | 46.8 | 41.8-51.8 | |
| Richer | 568 | 17.0 | 133 | 23.4 | 18.9-28.6 | 186 | 32.8 | 26.7-39.4 | 569 | 43.9 | 37.0-50.9 | |
| Richest | 469 | 14.1 | 226 | 48.3 | 39.5-57.2 | 109 | 23.4 | 18.0-29.7 | 132 | 28.3 | 22.5-35.0 | |
| **Community-level factors** | | | | | | | | | | | | |
| Region | | | | | | | | | | | | |
| North-Central | 488 | 14.6 | 111 | 22.9 | 19.1-27.2 | 142 | 29 | 24.6-34.4 | 233 | 47.9 | 42.0-53.8 | 0.05 |
| North-East | 594 | 17.8 | 152 | 25.6 | 21.3-30.5 | 167 | 28 | 24.4-32.1 | 275 | 46.2 | 41.4-51.2 | |
| North-West | 1261 | 37.8 | 349 | 27.7 | 24.4-31.2 | 408 | 32 | 29.1-35.8 | 503 | 39.9 | 36.5-43.4 | |
| South-East | 260 | 7.8 | 45 | 17.4 | 9.7-29.1 | 84 | 32 | 25.3-40.3 | 131 | 50.3 | 39.0-61.5 | |

*(Continued)*

**Table 2.** (*Continued*)

| | Study population | | Experiences of FI | | | | | | | | |
|---|---|---|---|---|---|---|---|---|---|---|---|
| | | | Food secure | | | Moderate FI | | | Severe FI | | |
| | Weighted | | Weighted | | | Weighted | | | Weighted | | |
| South-South | 361 | 10.8 | 93 | 25.9 | 17.7-36.2 | 101 | 28 | 22.4-34.6 | 166 | 46.1 | 37.5-54.9 | |
| South-West | 372 | 22.1 | 130 | 34.9 | 28.3-42.2 | 93 | 25 | 18.9-32.0 | 372 | 5.5 | 33.7-47.2 | |
| Place of residence | | | | | | | | | | | | |
| Rural | 2232 | 66.9 | 562 | 25.2 | 22.9-27.7 | 675 | 30 | 28.1-32.5 | 994 | 44.6 | 41.9-47.2 | 0.3 |
| Urban | 1105 | 33.1 | 319 | 28.9 | 24.3-34.0 | 321 | 29 | 25.2-33.1 | 465 | 42.1 | 37.2-47.1 | |
| Community-level poverty | | | | | | | | | | | | |
| Low | 1867 | 56.0 | 503 | 27.0 | 23.9-30.3 | 577 | 30.6 | 28.1-33.9 | 787 | 42.1 | 38.8-45.6 | 0.16 |
| Middle | 662 | 19.8 | 187 | 28.2 | 23.5-33.5 | 167 | 25.1 | 21.7-28.9 | 309 | 46.6 | 41.6-51.7 | |
| High | 807 | 24.2 | 191 | 23.7 | 19.8-28.1 | 252 | 31.3 | 26.7-35.1 | 364 | 45.0 | 40.5-49.6 | |

[a]*Notes*: Data are n (%). **All estimates are weighted for the survey's complex sampling design**

*p-Value < .05

**p-value < .01

***p-value < .001

$\chi^2$ = Denotes Rao-Scott Chi-Square

[†]May not total 100% due to missing values or rounding

evidence of multicollinearity. The variance inflation factor (VIF) values were all below the threshold value of 10.

Model building strategy.

To account for the categorical nature of our primary outcome variable, which comprises three levels displaying an intrinsic ordinal hierarchy, and that individual-level data is nested within higher-level categories (i.e. clusters), we fitted a series of two-level random intercept models to estimate the impact of individual- and community-level factors on food insecurity. We employed a generalized linear mixed model with a multinomial distribution and the CLOGIT link function to compute the cumulative odds for each category of food insecurity [25]. This approach enabled us to account for the violation of the independence assumption and avoid inflation of Type 1 errors. We used SAS PROC GLIMMIX with maximum likelihood with Laplace approximation (method = LAPLACE) and the CONTAIN method (DDFM = CONTAIN) to estimate the fixed effects as odds ratios (ORs) with 95% confidence intervals (CIs) for the multilevel logistic regression estimates. The models fitted include;

Null model: Model containing no predictors

Model I: Model containing only individual-level predictors

Model II: Model containing only community-level predictors

Model III: Model containing both individual- and community-level predictors.

The general equation of the random intercepts two-level multinomial logistic regression model used for analysis of predictors of FI takes the form

$$\text{Logit}(p_i) = \log[\pi_{ij}^{(s)}/\pi_{ij}^{(r)}] = \beta_0^{(s)} + \beta_1^{(s)}X_{1ij} + \beta_2^{(s)}X_{2ij} + \ldots + \beta_k^{(s)}X_{kij} + u_j^{(s)}, \ (\text{for } s = 1, 2, 3)$$

$\pi ij^{(s)}$ denotes the probability of FI, *s* (i.e. moderate FI = 2 or severe FI = 3) for woman *i*, in the *j*th community.

$\pi ij^{(r)}$ denotes the probability of being food secure (*r* = 1 for FI level used as reference category) for woman, *i*, in community, *j*

$\beta_0^{(s)}$ are the fixed regression intercepts for increased likelihood of FI, *s*.

$X_{(1-k)ij}$ are $1 - k$ explanatory variables defined at the individual or community level.

$\beta_{(1-k)}^{(s)}$ are the associated usual regression parameter estimates for being at risk of FI, *s*.

$uj^{(s)}$ are the community-level residuals for FI level, *s*. These are assumed to be normally distributed with mean zero and variance $\sigma^{2(s)}_u$. The community random effects may be correlated across food insecurity levels: covariance $(u_j^{(s2)}, u_j^{(s3)}) = \sigma^{(s2,3)}_u$, (s2 = moderate FI, s3 = severe FI).

We used a random intercept only model and calculated the intraclass correlation coefficient $(\tau_{00})$ by examining between-cluster variances and within-cluster variance. The within-cluster variance for logistic regression models is given by the variance of the standard logistic distribution. By using the logistic distribution variance of approximately 3.29 (or $\pi^2/3$), the ICC is calculated using the equation

$$\text{ICC} = [\tau_{00}/(\tau_{00} + 3.29)] * 100, \ \tau_{00} \text{ is the between}-\text{cluster variance.}$$

To evaluate how well the models accounted for cluster variability related to moderate and severe FI, we used the proportional change in variance (PCV) and compared the $\tau_{00}$ values of model I with the unconditional model $[\tau_{00(0)} - \tau_{00(n)}/ \tau_{00(0)}]$ and models II and III with the previous model constructed $[\tau_{00(n-1)} - \tau_{00(n)}/ \tau_{00(n-1)}]$. We compared different models using several measures of goodness of fit, including the -2 log likelihood, the Akaike information criterion (AIC), and the Bayesian information criterion (BIC). To ensure that parameter estimates are unbiased and consistent, we tested for the independence of irrelevant assumptions (IIA), using the Small-Hsiao test, which evaluates whether belonging to one FI category does not affect the other available categories [26].

## Results

### Characteristics of the sample of pregnant women

In terms of demographics, the pregnant women in the study had a mean (SE) age of 29.0±0.2 years and had given birth to a mean (SE) of 3.1±0.1 children. The average household size was 6.6±0.1, and there were 1.2±0.03 children under 5 years old in the household. Table 2 reveals that the vast majority of pregnant women were married and uninsured, with roughly one-fifth experiencing unintended pregnancies and nearly half coming from poor households. Most of the pregnant women lived in rural areas (66%) and were from the North-West geopolitical zone (37.8%). Additionally, more than half of the communities (clusters) had low socioeconomic status. Maternal age, parity, number of household members, and household wealth index all demonstrated a significant relationship with household FI (Table 2).

### Magnitude of household FI

As shown in Table 3, a substantial number of pregnant women experienced FI during the previous 12 months due to financial constraints. Specifically, over three-quarters of participants reported worrying about not having enough food, while a similar proportion were unable to afford healthy and nutritious food. In addition, a large proportion of participants reported consuming only a limited variety of foods (76%) and skipping meals due to lack of resources (67%). Furthermore, approximately 71% of pregnant women reported eating less than they thought they should because of financial constraints, while over 60% had run out of food at some point. Moreover, about one-half of pregnant women reported being hungry because they couldn't afford food, and nearly one-third of them had to go without eating for an entire day due to lack of resources. Based on the categorization of FI status, 26.4% (95% CI = 24.2–28.7) of pregnant women lived in households that were food secure, while 29.9% (95%

**Table 3. Status of FI Experience Scale (FIES) questions.**

| | Yes | | | No | | | |
|---|---|---|---|---|---|---|---|
| | *N* | % | 95% CI | *N* | % | 95% CI | $\chi^2$ |
| Worried you would not have enough food to eat because of a lack of money or other resources? | 2614 | 78.3 | 76.1–80.4 | 723 | 19.6 | 19.6–23.9 | 475.2*** |
| Unable to eat healthy and nutritious food because of a lack of money or other resources? | 2554 | 76.5 | 74.2–78.7 | 783 | 21.3 | 21.3–25.8 | 385.7*** |
| Ate only a few kinds of foods because of a lack of money or other resources? | 2550 | 76.4 | 74.1–78.6 | 787 | 23.6 | 21.4–25.9 | 387.3*** |
| Skip a meal because there was not enough money or other resources to get food? | 2222 | 66.6 | 64.3–68.9 | 1115 | 33.4 | 31.2–35.8 | 178.5*** |
| Ate less than you thought you should because of a lack of money or other resources? | 2365 | 70.9 | 27.0–31.4 | 972 | 29.1 | 68.9–73.0 | 280.5*** |
| Ran out of food because of a lack of money or other resources? | 2006 | 60.1 | 57.7–62.4 | 1331 | 39.9 | 37.6–42.3 | 68.6*** |
| Hungry but did not eat because there was not enough money or other resources for food? | 1700 | 51.0 | 48.5–53.4 | 1637 | 49.1 | 46.6–51.5 | 0.57 |
| Went without eating for a whole day because of a lack of money or other resources? | 1045 | 31.3 | 29.0–33.8 | 2292 | 68.7 | 66.2–71.0 | 203.7*** |

CI = 27.9–31.9) and 43.7% (95% CI = 41.3–46.1) lived in households with moderate and severe FI, respectively.

## Risk factors for FI: Multilevel analysis

Table 4 displays the results of the multilevel multinomial logistic regression analyses for FI levels. The fixed effects from the unconditional model represent the log odds of being at or below each food security level for pregnant women in a typical community. The results indicate that the log odds of being at or below severe levels of FI is -0.2674, corresponding to a predicted probability of 0.43. Similarly, the log odds of being at or below moderate levels of FI is 1.185, resulting in a cumulative probability of 0.77. Based on these, the predicted probability of being at the severe FI level in a typical community is 0.43, at the moderate FI level is 0.33, and at the food secure level is 0.23. The results also demonstrate significant variability across communities (clusters) in the likelihood of being at or below each FI level [$\tau_{00}$ = 0.6616, z(1317) = 6.69, $p$ >0.0001], indicating that these relationships vary significantly across communities. About 16.7% of the total variability in FI is accounted for by the communities, while the remaining 83.3% of the variability is due to unknown factors, or that are specific to individual pregnant respondents.

To determine the best fitting model for the data in this study, all four models were compared based on their fit. The results showed that the level-1 model (Model I) was a better fit than the unconditional model and Model II, as evidenced by changes in the AIC. Additionally, the combined level-1 and level-2 model (Model III) was found to fit better than the level-1 model (Model I). Based on these findings, we concluded that Model III is the best fitting model for the data used in this study. The final model (Model III) showed that several factors were significantly associated with moderate and severe FI. Pregnant women with more than five birth events had higher adjusted odds of being food-insecure (aOR = 1.38, 95% CI: 1.00–1.91, $p$ = 0.0497) compared to those with no birth events. Additionally, households with more than five members had significantly higher predicted adjusted odds of being food-insecure (aOR = 1.41, 95% CI: 1.15–1.71, $p$ = 0.001) than households with 5 or fewer members. Pregnant women living in urban areas had significantly higher odds of experiencing FI than those in rural areas (aOR = 1.57, 95% CI: 1.23–1.99, $p$ = 0.0003). However, as household wealth index increased, the predicted adjusted odds of being food-insecure decreased, with the poorest quintile having the highest odds and the richer and richest quintiles having significantly lower odds (all $p$<0.0001). Pregnant women in the North-West region had significantly lower odds of being food-insecure (aOR = 0.67, 95% CI: 0.51–0.88, $p$ = 0.004) compared to those in the North-Central region, while those in other regions did not differ significantly. Finally,

**Table 4. Multilevel multinomial logistic regression estimates of experiences of moderate and severe FI (versus food secure) for individual/household- and community-level factors[a].**

| Explanatory variables | Unconditional model | Model I | | Model II | | Model III | |
|---|---|---|---|---|---|---|---|
| **Fixed effects intercept[b]** | | | | | | | |
| Severe FI | -0.27(0.1) | -0.17(0.24) | | -0.22(0.12) | | 0.28(0.28) | |
| Moderate FI | 1.19(0.1) | 1.35(0.25) | | 1.24(0.12) | | 1.8(0.29) | |
| **Individual-/household-level factors** | | | | | | | |
| | | aOR [95% CI] | p-value | aOR [95% CI] | p-value | aOR [95% CI] | p-value |
| Age group (y) | | | | | | | |
| 15–24 (ref) | | 1.00 | | | | 1.00 | |
| 25–34 | | 1.15 [0.95–1.40] | 0.15 | | | 1.10 [0.90–1.35] | 0.33 |
| 35+ | | **1.31 [1.01–1.70]** | **0.04** | | | 1.21 [0.94–1.58] | 0.14 |
| Parity | | | | | | | |
| None (ref) | | 1.00 | | | | 1.00 | |
| 1–2 | | 1.18 [0.92–1.51 | 0.20 | | | 1.14 [0.89–1.47] | 0.30 |
| 3–4 | | 1.05 [0.79–1.40] | 0.75 | | | 1.00 [0.75–1.34] | 0.98 |
| ≥5 | | **1.40 [1.02–1.93]** | **0.04** | | | **1.38 [1.00–1.91]** | **0.0497** |
| Married/cohabiting | | | | | | | |
| No (ref) | | 1.00 | | | | 1.00 | |
| Yes | | 0.68 [0.44–1.05] | 0.08 | | | 0.79 [0.51–1.23] | 0.30 |
| Health insurance coverage | | | | | | | |
| Not insured (ref) | | 1.00 | | | | 1.00 | |
| Insured | | 0.66 [0.37–1.18] | 0.16 | | | 0.65 [0.36–1.16] | 0.14 |
| Pregnancy intention | | | | | | | |
| Unplanned (ref) | | 1.00 | | | | 1.00 | |
| Planned | | 0.98 [0.82–1.17] | 0.84 | | | 1.01 [0.85–1.21] | 0.92 |
| Household wealth index | | | | | | | |
| Poorest (ref) | | 1.00 | | | | 1.00 | |
| Poorer | | **0.77 [0.64–0.94]** | **0.01** | | | **0.64 [0.52–0.80]** | **< .0001** |
| Middle | | **0.70 [0.56–0.87]** | **0.0013** | | | **0.45 [0.34–0.59]** | **< .0001** |
| Richer | | **0.54 [0.43–0.70]** | **< .0001** | | | **0.29 [0.21–0.40]** | **< .0001** |
| Richest | | **0.27 [0.20–0.36]** | **< .0001** | | | **0.13 [0.09–0.19]** | **< .0001** |
| Household members | | | | | | | |
| ≤ 5 (ref) | | 1.00 | | | | 1.00 | |
| > 5 | | **1.31 [1.08–1.60]** | **0.01** | | | **1.41 [1.15–1.71]** | **0.001** |
| Religion | | | | | | | |
| Non-Christian (ref) | | 1.00 | | | | 1.00 | |
| Christian | | **1.59 [1.33–1.91]** | **< .0001** | | | 1.25 [0.99–1.59] | 0.07 |
| Children under 5 in household | | | | | | | |
| None (ref) | | 1.00 | | | | 1.00 | |
| 1 | | 1.02 [0.84–1.26] | 0.82 | | | 1.02 [0.32–1.25] | 0.85 |
| 2+ | | 1.03 [0.83–1.29] | 0.77 | | | 1.06 [0.84–1.33] | 0.62 |
| **Community-level factors** | | | | | | | |
| Region | | | | | | | |
| North-Central (ref) | | | | 1.00 | | 1.00 | |
| North-East | | | | 0.86 [0.67–1.11] | 0.25 | 0.79 [0.60–1.04] | 0.10 |
| North-West | | | | **0.73 [0.57–0.92]** | **0.01** | **0.67 [0.51–0.88]** | **0.004** |
| South-East | | | | 1.15 [0.82–1.61] | 0.43 | 1.14 [0.79–1.66] | 0.49 |
| South-South | | | | 1.23 [0.89–1.69] | 0.21 | 1.34 [0.95–1.91] | 0.10 |

(*Continued*)

**Table 4.** (Continued)

| Explanatory variables | Unconditional model | Model I | Model II | | Model III | |
|---|---|---|---|---|---|---|
| South-West | | | 0.79 [0.56–1.12] | 0.18 | 0.91 [0.63–1.31] | 0.62 |
| Place of residence | | | | | | |
| Rural (ref) | | | 1.00 | | 1.00 | |
| Urban | | | 0.93 [0.75–1.15] | 0.49 | **1.57 [1.23–1.99]** | **0.0003** |
| Community-level poverty | | | | | | |
| Low (ref) | | | 1.00 | | 1.00 | |
| middle | | | 1.14 [0.90–1.45] | 0.27 | **0.73 [0.56–0.95]** | **0.02** |
| High | | | **1.27 [1.01–1.60]** | **0.04** | **0.65 [0.49–0.87]** | **0.004** |
| **Random effects** | Unconditional model | Model I | Model II | | Model III | |
| Cluster-level variance (SE) | 0.66(0.1) | 0.69(0.1) | 0.64(0.1) | | 0.69(0.1) | |
| ICC (%) | 16.7 | 17.3 | 16.2 | | 17.4 | |
| Covariance | | | | | | |
| Log-likelihood | 7442.4 | 7269.7 | 7420.7 | | 7223.8 | |
| Explained variance (PCV, %) | | -4.6 | 7.8 | | -7.8 | |
| Model summary | | | | | | |
| AIC | 7448.4 | 7309.7 | 7442.7 | | 7279.8 | |
| BIC | 7464.0 | 7413.3 | 7499.8 | | 7425.0 | |

Abbreviations: SE = Standard Error; ICC = Intraclass correlation coefficient; AIC = Akaike's information criterion; BIC = Bayesian information criterion; PCV = Percentage Change in Variance

[a]*Notes*: **All estimates are weighted for the survey's complex sampling design**; Bolded text indicates statistical significance; Target: HFI; reference category: food secure; probability distribution: multinomial; link function cumulative logit.

[b] Estimates are presented as log odds

*$p < 0.05$

***$p < 0.0001$

***$p < 0.0001$

increasing levels of community-level poverty were associated with decreasing odds of moderate and severe FI, with areas having middle (aOR = 0.73, 95% CI: 0.56–0.95, $p = 0.02$) and high (aOR = 0.65, 95% CI: 0.49–0.87, $p = 0.004$) levels of poverty having significantly lower odds compared to areas with low poverty levels. The adjusted random effects, including the explained variance (represented by the proportional change in variance) of Models 0-III are also shown in Table 4. Relative to the null model, -4.6% of the variance in the risk of being food-insecure were explained by including level-1 predictors in the model. In the best fitting model (Model III), we observe that relative to Model II, -7.8% of the variability in FI were explained by including level-1 and level-2 predictors- in the full model. Based on the Small-Hsiao test of IIA assumption, we found no evidence to suggest violation of the IIA in the best fitting model (Mode III) as shown in Table 5.

**Table 5. Small-Hsiao test of IIA assumption for Model III.**

| | InL(full | InL(omit) | Chi Square | df | p-value | Hypothesis |
|---|---|---|---|---|---|---|
| Food Secure | -657.123 | -648.412 | 17.42 | 13 | 0.181 | For Ho |
| Moderate FI | -760.412 | -753.269 | 14.29 | 13 | 0.354 | For Ho |
| Severe FI | -884.621 | -881.341 | 6.56 | 13 | 0.923 | For Ho |

## Discussion

Food insecurity remains a public health challenge especially in low- and middle-income countries (LMICs), with pregnant women being particularly vulnerable. To gain better insight regarding the extent of this issue in Nigeria, we investigated the burden of FI and its contributing factors among pregnant women, using data from the 2021 Nigeria MICS6 survey. Our findings revealed a high prevalence of FI among pregnant women in Nigeria, with 73.6% of women aged 15–49 years being food-insecure with about 44.0% experiencing severe FI. After adjusting for several potential confounders, our multilevel analysis based on the best fitting model (Model III) revealed significant associations between several factors and FI among pregnant women in Nigeria.

At the individual- and household-levels, we found that having at least 5 or more birth events, household wealth index and living in households with more than five members were all associated were associated with higher odds of pregnant women experiencing FI, after adjusting for confounders. Specifically, we observed that both higher parity (5 or more) and residing in a household with more than five member were positively and significantly associated with FI. Having more children and a larger household size can increase the household's food needs and expenses, making it more challenging to afford and access adequate amounts of nutritious food. Moreover, higher parity and larger household size can result in insufficient energy and nutrient intake, contributing to FI and poor health outcomes for individuals. To address high parity and large household size, expanding access to family planning services may help reduce FI and improve maternal and child health outcomes. Evidence from Tanzania shows that women exposed to household hunger were significantly less likely to have further desire for childbearing compared to their counterparts not experiencing household hunger (aOR = 0.8, 95% CI: 0.69–0.96) [27]. However, another study in Ethiopia showed low uptake of modern contraceptive methods among women of reproductive age in food-insecure households compared to food-secure households (aOR = 1.69, 95% CI: 1.03, 2.66) [28].

Also, we observed that the odds of experiencing FI were highest among women residing in the poorest households. However, as household wealth index increased, pregnant women were less likely to experience FI. This may reflect the role that financial resources play in ensuring access to adequate amounts of nutritious food and that living in areas with more economic opportunities may help alleviate FI. Numerous factors, including a lower socioeconomic status, employment in roles with diminished wages and reduced work hours, and involvement in unpaid domestic responsibilities, have been linked, to some extent, to the gendered dimensions of FI [29]. One study in Nigeria showed that male headed households were more likely to be food secure compared to female headed households [30]. Implicitly, these findings underscore the significance of adopting gender-centered approaches as critical window of opportunity for government and policy stakeholders to address and alleviate the repercussions of FI on maternal, reproductive, and pregnancy outcomes.

In terms of community-level factors, we found that several factors were associated with experiences of FI. With respect to place of residence, we observed that pregnant women residing in urban areas compared to their counterparts in rural areas were more likely to report FI. Though counterintuitive, this could be due to the fact that pregnant women living in urban households with restricted access to affordable and healthy food may encounter difficulties in obtaining sufficient amounts of nutritious food and as a result experience FI. Furthermore, the high reliance on purchased food and the relatively high cost of food in urban areas may contribute to a higher risk of FI for low-income urban households than in rural areas [31]. In addition, the disruptions in food supply chain systems, particularly pronounced in urban regions with restricted agricultural capacities, notably during the COVID-19 pandemic [32], might

have contributed to the disparities in FI between rural and urban settings among the pregnant women in our study. Our finding however, contrasts with findings in a study by Rutayisire et al. [13] which revealed that pregnant women in rural areas were significantly more likely to experience FI.

Similar to our findings with household wealth index, we also observed that pregnant women residing in communities with low poverty levels compared to those living in communities with average and high poverty levels were more likely to experience FI. Also, our findings revealed that residing in the North Western region of Nigeria relative to the North Central region had a protective effect on FI among pregnant women. Although it is postulated that the prevalence of droughts and floods in the North Eastern and North Western parts of Nigeria are factors which may predispose households to FI [33], It is not clear how pregnant women in the North West are shielded from FI. However, a plausible explanation could be that North Western region might have more favorable agro-ecological conditions which allow for increased agricultural production and a more stable local food supply. Furthermore, it is possible that this region might have strong local food systems, ensuring a more consistent supply of food through local markets and production.

Our study therefore expands on the current understanding of FI among pregnant women in Nigeria, as there have been few investigations in this area. A previous study conducted in Ogun state found that 46.4% of pregnant women lived in food-insecure households [14], which is substantially lower than the prevalence identified in our study. There are several possible reasons for these differences in FI prevalence. Firstly, our study utilized data from a nationally representative sample of pregnant women, providing a larger sample size compared to the previous study. Additionally, the measures of FI used in the two studies were different; while the previous study used a short-form Food Security Survey, our study used the FI Experience Scale (FIES), which is recommended by the Food and Agriculture Organization (FAO) for consistent comparison of FI trends both within Nigeria and with other countries. This is important for monitoring progress in reducing FI among pregnant women and achieving the Sustainable Development Goals (SDGs). Furthermore, the data used in our study was based on a survey conducted during a time when Nigeria, along with many other countries, was recovering from the impact of the COVID-19 pandemic. It is possible that the pandemic may have influenced food security in households, but the extent of this impact is not yet understood. Studies of FI among pregnant women in other parts of sub-Saharan Africa have reported prevalence rates of 66.7% in Malawi [34], 53.1% in Rwanda [13], 42% in South Africa [35], 77.5% in Ghana [36], 49.4% in Ethiopia [37], and 76.5% among peripartum HIV women in Kenya [38]. While there is considerable variation in these estimates, probably due in part to differences in measurement of FI and study sample sizes, there is clear evidence indicating that FI remains a significant issue for pregnant women in sub-Saharan Africa.

The findings from our study have Implications for research, policy and practice. Future research should investigate the disparities in pregnancy and postpartum outcomes are shaped by maternal experiences of preconception and prenatal FI. Additionally, given the high prevalence of FI among pregnant women in Nigeria, it may be worthwhile for women's health providers to adopt strategies for integrating antenatal screening for FI into routine prenatal care. This screening approach can help identify pregnant women who are currently experiencing or are at risk of FI and facilitate their referral to community-based FI interventions. Lastly, a study conducted in the United States revealed that pregnant women experiencing FI were more likely to delay or forgo health care due to cost compared to their food secure counterparts [39]. Given this insight, there is a need to pragmatically support the formulation and implementation of effective policies that will ensure pregnant women in Nigeria, despite experiencing FI, do not have unmet health care needs, which potentially could influence pregnancy outcomes.

### Strengths and limitations

Our study has several strengths that contribute to its significance and relevance in addressing the issue of FI among pregnant women. Firstly, it fills the gaps in the limited available evidence on FI among pregnant women. By providing new insights into the prevalence and associated factors of FI among this vulnerable population, the findings from our study can inform the development of evidence-based interventions that address their unique challenges. Furthermore, the use of data from a nationally representative survey in our study enhances the generalizability of its findings. The estimates generated from this study can be applied to the entire Nigerian population of pregnant women, providing a more accurate representation of the prevalence of FI compared to previous studies that were limited to specific geographic areas or populations. Lastly, the multilevel approach employed in our study corrected for the violation of the independence assumption and minimized the type 1 error rate, ultimately strengthening its internal validity. This approach allowed for a more comprehensive examination of the complex relationships between FI and its associated factors, contributing to a more nuanced understanding of the issue.

Despite the strengths of our study, there were some limitations that need to be acknowledged. First, we were unable to adjust for the impact of the COVID-19 pandemic on FI among pregnant women. This could have affected the estimates of FI prevalence and associated factors in our study. Also, the cross-sectional design of the study limits our ability to establish causality and the temporal relationship between FI and its associated factors. Lastly, assessment of FI was based on participant's recall which is subject to recall bias. By implication, the prevalence of FI in our study could be underestimated or overestimated due to inaccuracies in participant recall. Nevertheless, our study provides valuable insights into the prevalence and associated factors of FI among pregnant women in Nigeria, which can inform the development of evidence-based interventions to address this critical public health issue.

## Conclusions

The detrimental effects of FI on maternal and child health are well-established. Through our study, we have provided evidence of a substantial burden of FI among pregnant women in Nigeria and have identified associated factors. Our findings underscore the urgent need for multilevel interventions that target FI as a crucial social determinant of health and well-being for pregnant women. Such interventions can potentially improve health outcomes and reduce health disparities among this vulnerable population.

## Author Contributions

**Conceptualization:** Otobo I. Ujah, Joseph-Anejo Okopi.

**Data curation:** Otobo I. Ujah.

**Formal analysis:** Otobo I. Ujah, Chukwuemeka E. Ogbu, Joseph-Anejo Okopi, Russell S. Kirby.

**Investigation:** Otobo I. Ujah.

**Methodology:** Otobo I. Ujah, Chukwuemeka E. Ogbu, Russell S. Kirby.

**Software:** Chukwuemeka E. Ogbu.

**Supervision:** Otobo I. Ujah, Joseph-Anejo Okopi, Russell S. Kirby.

**Validation:** Russell S. Kirby.

**Writing – original draft:** Otobo I. Ujah, Pelumi Olaore, Chukwuemeka E. Ogbu, Joseph-Anejo Okopi, Russell S. Kirby.

**Writing – review & editing:** Otobo I. Ujah, Pelumi Olaore, Chukwuemeka E. Ogbu, Russell S. Kirby.

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
