## [Decision Letter · Decision Letter 0]

8 Aug 2023

PGPH-D-23-01116

Prevalence and determinants of food insecurity among pregnant women in Nigeria: a multilevel mixed effects analysis

Dear Dr. Ujah,

Thank you for submitting your manuscript to PLOS Global Public Health. After careful consideration, we feel that it has merit but does not fully meet PLOS Global Public Health’s publication criteria as it currently stands. Therefore, we invite you to submit a revised version of the manuscript that addresses the points raised during the review process.

We look forward to receiving your revised manuscript.

Kind regards,

Sonali Sarkar

Academic Editor

Journal Requirements:

Additional Editor Comments (if provided):

Reviewers' comments:

Reviewer's Responses to Questions

**Comments to the Author**

1. Does this manuscript meet PLOS Global Public Health’s publication criteria? Is the manuscript technically sound, and do the data support the conclusions? The manuscript must describe methodologically and ethically rigorous research with conclusions that are appropriately drawn based on the data presented.

Reviewer #1: Yes

Reviewer #2: Yes

Reviewer #3: Yes

2. Has the statistical analysis been performed appropriately and rigorously?

Reviewer #1: Yes

Reviewer #2: Yes

Reviewer #3: Yes

3. Have the authors made all data underlying the findings in their manuscript fully available (please refer to the Data Availability Statement at the start of the manuscript PDF file)?

Reviewer #1: Yes

Reviewer #2: Yes

Reviewer #3: Yes

4. Is the manuscript presented in an intelligible fashion and written in standard English?

Reviewer #1: Yes

Reviewer #2: Yes

Reviewer #3: Yes

5. Review Comments to the Author

Reviewer #1: Overall, this study investigating the prevalence of household food insecurity and its association with individual- and contextual-level factors among pregnant women in Nigeria is both timely and important, given the potential negative impacts of food insecurity on maternal and child health. The methodology appears appropriate, utilizing a cross-sectional design based on the 2021 Nigerian Multiple Indicator Cluster Survey (MICS6) data and applying weighted multilevel multinomial logistic regression models to analyze the associations.

Reviewer #2: This manuscript from secondary data analysis of publicly available data majorly adhere to analysis and reporting of national level representative complex survey data.

The following clarifications/suggestions are recommended:

What all the factors being considered while estimating weighted %. (i.e Out of the factors such as stratification, clustering and Non Response rate, weightage mentioned in this manuscript refers to what?)

Does the survey design include more than one pregnant women in a family? In that case, the nested multi level modelling has to address the household level factors as well before adjusting for community level factors?

As the community level poverty index is in turn derived from individual household from the same community cluster, there is a possibility of multi collinearity? Did authors check for any collinearity while developing the model?

As given in the manuscript, dependent variable has ordinal categories. In this case the reason for using multinomial regression could be explained. While developing multinomial regression model, how did the assumptions of independent irrelevant alternatives were followed?

What is the extent of missing data on food insecurity measures among eligible pregnant women?

To understand how the geographical location (region) has affected food insecurity, it is better to describe the contextual factors (especially those having high food insecurity) associated with these regions

Similar to health insurance coverage, access to other social supportive measures (food security programs) and coverage for ANC services would also have influenced the food security level (The current model does not include these factors)

As reported in previous literatures gender dimensions and cultural norms play a major role in food security measures especially among women. This survey data directly may/may not have these data. However, speculations under these dimensions in discussion could enhance the understanding.

Reviewer #3: Abstract

1. Authors need to emphasise on the problem statement at the introduction section

2. Include the sample size

Introduction

Authors should be specific on the interventions and programs and list some of the efforts and developmental achievement.

The gap for the study is not clear. They should strengthen it.

There should be a theory guiding the study

Methodology

Why is marital status categorised as “yes” and “no”? What does yes and no mean

They should include education of mother, women empowerment questions and some household and paternal characteristics

Results and discussion

They should link the theory to the discussions

The recommendation is not strong. They should strengthen it

6. PLOS authors have the option to publish the peer review history of their article (what does this mean?). If published, this will include your full peer review and any attached files.

**Do you want your identity to be public for this peer review?** For information about this choice, including consent withdrawal, please see our Privacy Policy.

Reviewer #1: **Yes: **Tanveer Rehman

Reviewer #2: **Yes: **Kalaiselvi S

Reviewer #3: No

---

## [Editor Report · Decision Letter 1]

27 Sep 2023

Prevalence and determinants of food insecurity among pregnant women in Nigeria: a multilevel mixed effects analysis

PGPH-D-23-01116R1

Dear Dr Ujah,

We are pleased to inform you that your manuscript 'Prevalence and determinants of food insecurity among pregnant women in Nigeria: a multilevel mixed effects analysis' has been provisionally accepted for publication in PLOS Global Public Health.

Best regards,

Sonali Sarkar

Academic Editor

The changes made to the manuscript are as per the suggestions of the reviewers and are found to be satisfactory. We thank the authors for incorporating all the suggested changes.